# Multiple Dosing and Preactivation of Mesenchymal Stromal Cells Enhance Efficacy in Established Pneumonia Induced by Antimicrobial-Resistant *Klebsiella pneumoniae* in Rodents

**DOI:** 10.3390/ijms24098055

**Published:** 2023-04-29

**Authors:** Declan Byrnes, Claire H. Masterson, Hector E. Gonzales, Sean D. McCarthy, Daniel P. O’Toole, John G. Laffey

**Affiliations:** 1Anaesthesia, School of Medicine, University of Galway, H91 TR33 Galway, Ireland; 2Regenerative Medicine Institute (REMEDI) at CÚRAM Centre for Research in Medical Devices, Biomedical Sciences Building, University of Galway, H91 TR33 Galway, Ireland; 3Department of Anaesthesia, Galway University Hospitals, SAOLTA University Health Group, H91 YR71 Galway, Ireland

**Keywords:** mesenchymal stem cells, pneumonia, inflammation, preactivation, multiple doses

## Abstract

Antimicrobial-resistant (AMR) bacteria, such as *Klebsiella* species, are an increasingly common cause of hospital-acquired pneumonia, resulting in high mortality and morbidity. Harnessing the host immune response to AMR bacterial infection using mesenchymal stem cells (MSCs) is a promising approach to bypass bacterial AMR mechanisms. The administration of single doses of naïve MSCs to ARDS clinical trial patient cohorts has been shown to be safe, although efficacy is unclear. The study tested whether repeated MSC dosing and/or preactivation, would attenuate AMR *Klebsiella pneumonia*-induced established pneumonia. Rat models of established *K. pneumoniae*-induced pneumonia were randomised to receive intravenous naïve or cytomix-preactivated umbilical cord MSCs as a single dose at 24 h post pneumonia induction with or without a subsequent dose at 48 h. Physiological indices, bronchoalveolar lavage (BAL), and tissues were obtained at 72 h post pneumonia induction. A single dose of naïve MSCs was largely ineffective, whereas two doses of MSCs were effective in attenuating *Klebsiella* pneumosepsis, improving lung compliance and oxygenation, while reducing bacteria and injury in the lung. Cytomix-preactivated MSCs were superior to naïve MSCs. BAL neutrophil counts and activation were reduced, and apoptosis increased. MSC therapy reduced cytotoxic BAL T cells, and increased CD4^+^/CD8^+^ ratios. Systemically, granulocytes, classical monocytes, and the CD4^+^/CD8^+^ ratio were reduced, and nonclassical monocytes were increased. Repeated doses of MSCs—particularly preactivated MSCs—enhance their therapeutic potential in a clinically relevant model of established AMR *K. pneumoniae*-induced pneumosepsis.

## 1. Introduction

Pneumonia induced by antimicrobial-resistant (AMR) bacteria is a highly morbid condition. Our ageing patient population has high rates of immunosuppression (cancer, transplant, and immune disorders) and is at high risk for infection by AMR bacterial species, especially when in hospital environments. Hospital-acquired pneumonia (HAP) occurs in ~0.5–2.0% of all hospitalised patients, with mortality rates of 30–70% reported [1]. *Klebsiella pneumoniae* accounts for 3–8% of these nosocomial bacterial infections and 3–5% of all community-acquired pneumonia in developed countries [2]. The ability of *K. pneumoniae* to easily acquire multiple resistance genes and virulence factors highlights the urgency in addressing this [3], and efforts are being made to focus on the ‘silent pandemic of AMR’ by the World Health Organisation (WHO) [4]. Initially, following infection, patients develop an acute inflammatory response which can either become uncontrolled leading to early death or develop into persistent inflammation leading to immune cell exhaustion or paralysis [5]. This leaves the patient susceptible to secondary infection [6]. 

Mesenchymal stromal cells (MSCs) can favourably modulate the immune response to pneumonia, reducing the potentially deleterious proinflammatory response, while enhancing bacterial killing by host immune cells [7]. MSC therapy can also enhance resolution and repair following injury, as demonstrated following ventilator-induced lung injury [8], where even delayed therapy demonstrated ongoing, albeit reduced, effectiveness at later timepoints. However, preclinical studies in infection/sepsis models to date have generally focused on the earliest phases of the infection process. In the clinical setting, MSC administration would only be feasible significantly later in the infection process, limiting the clinical translatability of preclinical studies to date. A second issue is that clinical studies have so far used single doses of naïve MSCs. These issues may partly explain the lack of a clear efficacy signal in clinical studies of acute respiratory distress syndrome (ARDS) [9] and sepsis [10,11] to date. 

Several attempts have been made to enhance the therapeutic efficacy of MSCs to increase effect without the need to administer large cell numbers. Multiple strategies have been employed for this including genetic modification, environmental stressors, and exposure to proinflammatory cytokines [12]. We previously demonstrated the enhanced therapeutic properties of MSCs after cytokine preactivation when administered early following *K. pneumoniae* infection [13]. Here, our hypothesis is that, when given at a later timepoint, the MSC therapy is less effective, but this can be rescued by administration of preactivated MSCs. 

In these studies, we wished to determine the potential for repeated doses of MSCs and for MSC preactivation to enhance therapeutic efficacy, when administered later in the course of infection induced by an AMR strain of *K. pneumoniae*. We hypothesised that multiple dosing would prove more effective than single dosing, and that preactivation would enhance efficacy in this model of established AMR bacteria-induced pneumonia. If our hypotheses are proven correct, this approach would yield a directly clinically translatable therapeutic approach for patients with AMR-induced established clinical pneumonia.

## 2. Results

### 2.1. Repeated Dosing with Preactivated MSCs Restored Lung Function and Reduced Bacterial Load

MSC therapy restored lung function and reduced *K. pneumonia* bacterial counts in established infection. While naïve MSCs were less effective, a single dose of cytomix-licensed MSCs at 24 h was effective, with a repeat dose at 48 h further improving these parameters (Figure 1). Cytomix-licensed MSC (PA-MSC) administration improved arterial oxygenation after one and two doses (Figure 1A); however, only a repeated dose of PA-MSCs improved the alveolar–arterial oxygen gradient (AaDO_2_, Figure 1B). Both single and repeated doses of PA-MSCs, but only a repeated dose of naïve MSCs (MSC), significantly improved static lung compliance and blood lactate levels (Figure 1C,D). Both single and repeated doses of MSCs and PA-MSCs significantly reduced BAL bacteria counts, with a single or double dose of PA-MSCs significantly reducing circulating bacteria (Figure 1E,F). 

### 2.2. Preactivated MSCs Resolve Injury and Restore Lung Structure

Histological analysis showed that *K. pneumonia* infection caused a severe morphological lung injury and reduced airspace fraction compared to sham animals (Figure 2A). Administration of either a single dose or repeated dose of PA-MSCs significantly improved the airspace-to-tissue fraction (Figure 2A,E,G). Representative images of sham, vehicle, single dose naïve, single dose preactivated, repeated dose naïve, and repeated dose preactivated MSCs (Figure 2B–G) clearly showed significant alveolar damage after the infection, as well as the potential for PA-MSC therapy to resolve injury and restore lung structure.

### 2.3. Inflammatory Cytokines and Chemotaxis Agents Were Reduced in the BAL of Preactivated MSC-Treated Animals

Both single and repeated PA-MSC administration significantly reduced the inflammatory cytokines IL-6, CINC1, and MIP-3α (Figure 3A–C), while only repeated doses of PA-MSCs significantly reduced MCP-1 (Figure 3D). Both single and repeated dosing of MSCs significantly reduced IL-6, with the repeated dose significantly reducing CINC1. PA-MSC dosing and repeated MSC administration significantly reduced MMP-9 (Figure 3E). The administration of MSCs to our model of established infection did not reduce the inflammatory cytokines TNF-α, IL-1β, or IFN-γ (Appendix A).

### 2.4. MSCs Reduce BAL White Blood Cells and Alter Neutrophil and Monocyte/Macrophage Profiles

*K. pneumoniae* infection increased alveolar WBC infiltration, consisting of macrophage/monocytes and neutrophils (Figure 4A–C). Neutrophils were significantly decreased after single and double doses of PA-MSCs and a double dose of MSCs (Figure 4B). There was no significant change between MSC treatment and vehicle control groups for the total number of macrophage/monocytes in the BAL (Figure 4C) 

Repeated doses of MSC and PA-MSCs significantly increased neutrophil apoptosis in the BAL (Figure 4D). The activation state of the neutrophils, as denoted by CD11b/c expression, was significantly reduced after PA-MSC—but not MSC—administration (Figure 4E). There was no change to neutrophil cell function in terms of phagocytosis and superoxide anion production (Appendix A). There was also no reduction in the apoptosis rate of the cells other than neutrophils in the BAL, which included monocytes/macrophages (Appendix A), and the activation level of these cells was increased with a double dose of MSCs and PA-MSCs (Appendix A). 

Administration of MSCs significantly increased the phagocytic index of BAL macrophages (Figure 4F). Single and double doses of MSCs, as well as a double dose of PA-MSCs, significantly increased the percentage of superoxide anion-positive cells in the BAL (Figure 4G). Sham data for neutrophil apoptosis and activation state were not obtainable due to the low number of WBCs and low levels of neutrophils present in the BAL.

### 2.5. MSC Therapy Alters the Activation of Infiltrating Alveolar T Cells

The reduction in BAL WCC was partially accounted for by a significantly reduced infiltration of T cells into the BAL of our animals (Figure 5A), regardless of MSC activation state or dosing regimen. Flow cytometry analysis showed that there was no change amongst all treatment groups for the total number of T helper cells in the BAL (Figure 5B). Repeated doses of PA-MSCs increased the percentage of CD4^+^ T cells expressing the activation and exhaustion regulation marker PD-1 (Figure 5C). However, the surface expression level per cell remained unchanged, as did the proliferation rate of these cells (Appendix A). After single and repeated doses of MSCs, there was an increased proportion of CD25-expressing T helper cells, indicative of an enhanced activation state and the potential of increased T regulatory cells present in the lung (Figure 5D). The cell-surface expression level of CD25 per cell remained unchanged across all the samples (Appendix A). 

There was a significant reduction in CD8^+^ cytotoxic T cells in the BAL across all treatment groups (Figure 5E), accounting for the total reduction in T cells in the BAL. The activation state of these cells, as shown by an increased percentage of the cells expressing PD-1, was significantly increased after single and repeated doses of PA-MSCs and repeated doses of MSCs (Figure 5F). There was no change in the surface marker expression level of PD-1, CD25, CD38, or proliferation rate of these cells, but the total number of CD8^+^ CD25^+^ cells was significantly reduced by a single dose of PA-MSCs (Appendix A). There was a significant increase in the ratio of BAL CD4/CD8 T cells across all treatment groups (Figure 5G). Sham data were not obtainable due to the low number of WBC and low levels of T cells present in the BAL.

### 2.6. MSCs Restore Systemic Haematological Profile and Reduces Circulating Inflammatory Cytokines 

*K. pneumoniae* infection significantly increased the blood counts of RBCs, haemoglobin, haematocrit, and platelets as determined by a haemoanalyser (Appendix A). MSC therapy significantly decreased platelets and increased mean corpuscular haemoglobin compared to vehicle (Appendix A) with PA-MSC and repeated doses of MSCs significantly reducing total platelets back to healthy control levels (Appendix A). 

There was no observable change in the serum concentrations of MMP-9 and MIP-3α (Appendix A) after MSC administrations. However, a single dose of MSC increased the level of MCP-1 compared to vehicle (Appendix A), and both single and double doses of PA-MSCs significantly reduced serum concentrations of IL-1β (Appendix A). 

### 2.7. MSCs Alter Systemic CD4^+^ T Cells, but Not Cytotoxic CD8^+^ T Cells

*K. pneumoniae* infection induced a significant increase in systemic white blood cells and granulocytes in the untreated control groups, but no significant change in the MSC-treated groups compared to sham (Appendix A). Circulating T cell and lymphocyte numbers remained unchanged across all groups (Appendix A). Repeated administration of PA-MSCs significantly increased total monocytes in circulation compared to vehicle (Appendix A). Although the absolute number of CD4^+^ T cells did not change, (Appendix A), the percentage of CD4^+^ CD25^+^ T cells was significantly increased by the administration of single and double doses of MSCs, and a double dose of PA-MSCs, without exceeding sham control levels (Appendix A). A single dose of MSCs at 24 h post infection was able to significantly decrease the total CD8^+^ T cell number, with the other MSC-treated groups in close range (Appendix A). There was a slight increase in the ratio of CD4^+^ to CD8^+^ T cells in circulation after MSC administration with a single dose of cytomix, reaching significance (Appendix A).

### 2.8. MSC Administration Alters Systemic Neutrophil Activation, but Not Proportion or Function

While there was an increase in the total number of circulating neutrophils (Appendix A), there was no significant difference across any of the treatment groups for systemic neutrophil apoptosis (Appendix A). Single and repeated doses of naïve MSCs and repeated doses of cytomix-licensed MSCs significantly reduced the surface activation marker CD11b/c (Appendix A). Neutrophil function was unchanged after MSC therapy as there was no change to their phagocytosis (Appendix A). Sham data were not obtainable for apoptosis, CD11b/c surface level expression, or phagocytosis as these are presented as relative changes, with shams not being available for every datapoint.

## 3. Discussion

Pneumonia induced by antimicrobial-resistant (AMR) bacteria is a serious respiratory illness with a high mortality rate in patients admitted to the ICU [1]. Common causative bacterial pathogens include the opportunistic Gram-negative bacterium *K. pneumoniae*, which has been implicated in both hospital-acquired and ventilator-associated cases of pneumonia [14,15]. *K. pneumoniae*’s ability to acquire multiple antimicrobial resistance and virulence factor genes has led to a significant increase in AMR genes being revealed in its genome [3]. Recently, a correlation between highly antimicrobial-resistant (AMR) bacteria and increasing mortality was indicated [16], which is concerning given the considerable resultant healthcare service impact and global economic burden [17]. Accordingly, the World Health Organisation’s ‘One Health’ joint plan of action for 2022–2024 outlines action on ‘curbing the silent pandemic of AMR’ [4].

MSCs have considerable promise as a potential treatment for bacterial pneumonia due to their immune modulation capacity [18], which extends to AMR strains [13]. The immunoevasive properties of MSCs have been well studied and documented, which allows for allogenic administered without adverse reactions, with xenogeneic administration of human MSCs into animals modelling the extreme end of this spectrum for host immune responses against nonself cells [19,20]. However, recent studies questioned this paradigm by showing a humoral response against human MSCs given to healthy mice, showing increased immunoglobulin production 10 days after the administration [21] while others showed increased circulatory CD4^+^ T cell numbers and increased serum IL-6 levels [22]. These factors were taken into consideration for data interpretation to ensure that the effects seen were due to antisepsis responses and not anti-MSC responses.

The efficacy of MSCs in preclinical studies has yet to be demonstrated in clinical trials [23] in patients with ARDS [9] or sepsis [10,11]. This ‘translational failure’ could be due to several factors including requirement for testing in more clinically relevant preclinical models, a more potent cell treatment, and/or better dosing regimens. We demonstrated in our previous study that the effects of naïve MSCs can be enhanced using cytomix activation in our preclinical pneumonia model, with a single MSC administration in presymptomatic animals, 1 h after bacterial infection [13]. 

Here, we altered the dosing to more clinically relevant timepoints, i.e., during established infection, followed by a repeat dose 24 h later. In this established, intermediate-stage pneumonia model, the effects of both MSC preactivation and serial dosing were revealed. 

A single dose of naïve MSCs during the established infection phase was demonstrated to be largely ineffective in the vast majority of parameters examined in this study. Conversely, the administration of multiple doses of preactivated MSCs enhanced the resolution of established AMR pneumonia. By administering cytomix-licensed MSCs 24 and 48 h after the bacterial inoculation, lung function was restored, showing significantly improved arterial oxygenation and alveolar–arterial oxygen gradient. There was also a significant reduction in lung oedema with cytomix-licensed MSC administration that did not occur when administered during the acute phase of infection, as previously demonstrated [13]. This shows the importance of the therapeutic window for MSCs, whereby they can enhance the resolution of infection when administered before or during the peak of infection. However, MSCs are typically short-lived [24,25,26] and may require multiple administrations [27]. 

There was a significant resolution of diffuse alveolar damage and atelectasis after cytomix-licensed MSC administration, showing that preactivated MSCs can restore lung structure even in the setting of established untreated infection. This is a promising result of MSC therapy and would contribute to positive long-term outcomes in ARDS patients who can suffer the ongoing effects of lung fibrosis and poor oxygen diffusion impairment (reviewed in [28]). 

ARDS is characterised by an acute inflammatory phase in which elevated levels of cytokines are found in the BAL and blood of patients [29], which contributes to ongoing inflammation and tissue damage in this phase. Here, it was demonstrated that administration of MSCs during established infection did not alter the potent earlier-phase proinflammatory cytokines TNF-α, IL-1β, and IFN-γ in the BAL, but there was a significant reduction in BAL MMP-9, IL-6, CINC1, MIP-3α, and MCP-1, which are considered to be functional later-phase cytokines with downstream effects on the innate and adaptive immune systems. CINC1 and MIP-3α (at high concentrations) are potent neutrophil chemoattractants [30,31], while MMP-9 is an inflammatory ECM degrader and cytokine activator produced by neutrophils [32], and their reduction likely accounted for the decreased neutrophil infiltration into the alveolar airspace demonstrated here. This, coupled with an increased BAL neutrophil proapoptotic phenotype, resulted in their enhanced clearance from the alveolar airspace and could account for the improved lung physiology shown [33]. There was an increase on average in circulating neutrophils which had a reduced integrin activation state after MSC treatment, indicating reduced infiltration, without any functional difference in terms of apoptosis, phagocytosis, and superoxide anion production. The infection-induced reduction in functionality of the macrophages and monocytes in the BAL was restored after single and multiple naïve MSC treatments. There was, however, no increase in macrophage/monocytes in the BAL, which may have been due to the rapid depletion in alveolar macrophages over the course of the infection, as previously reported [34], likely occurring before the MSC administration could confer protective effects [35]. 

In sepsis and ARDS, T cells can become dysfunctional, leading to their inactivation, enhanced apoptosis, and reduced responsiveness to stimuli [36]. During established *K. pneumoniae* pulmonary infection we observed a large increase in T cell infiltration to the lung airspace. After MSC treatment there was a significant reduction in the total number of T cells in the BAL which correlated with a significant reduction in cytotoxic CD8^+^ T cells. The CD4^+^/CD8^+^ ratio was similar to that demonstrated after MSC administration during early infection [13]. During intermediate or established infection, T cell exhaustion and immune cell paralysis can be initiated. This is characterised by a reduction in proliferative capacity, increased rate of apoptosis, and increased expression of the activation and inhibitory molecule PD-1 on the cell surface [37]. While repeated administrations of cytomix-licensed MSCs significantly reduced the presentation of exhaustion and inactivation indices in both CD4^+^ and CD8^+^ T cells, further studies are needed to establish whether this remains the case in a chronic infection, where exhaustion occurs [37], with respect to the functionality of these cells from the animals after MSC administration.

Cytomix-licensed MSCs increased the CD4/CD8 ratio of T cells in circulation, which was seen by a decrease, on average, in the total number of circulating CD8^+^ T cells, showing no xenogeneic response to the human MSCs in this parameter. This typically indicates a stronger immune system, as low ratios are seen in sepsis and HIV-positive patients [38]. Upper quartile ratios of CD4^+^ to CD8^+^ have been implicated in higher COVID-19 mortality rates during the acute phase due to lower CD8^+^ T cell expansion [39]. Here, the majority of treatment groups remained on par with healthy controls; thus, it can be safely assumed that impaired CD8^+^ T cell expansion is not an issue. CD38 was also unchanged, which is common for patients, as the early phase of sepsis has minimal activation of circulating CD8^+^ T cells or an increased expression level of CD38 [40]. In this series, the percentage of circulatory CD25-co-expressing CD4^+^ T cells was not elevated between treatments and healthy controls, while the percentage of CD25-co-expressing CD8^+^ T cells was unchanged between vehicle and MSC treatments. Elevated levels of CD25 co-expression have been implicated in GvHD rejection, which was not seen here [41].

Naïve MSC administration and repeated doses of cytomix-licensed MSCs significantly enhanced the number of CD25 expressing CD4^+^ T cells in the BAL, indicating improved function for controlling immune cell responses [42]. However, it needs to be further investigated whether this shows an increase in the number of activated T helper cells or an increase in the number of T regulatory cells present in the lung, which both express CD25. Circulating cells, which were CD11b/c^+^, had an increased surface level expression of the marker, indicative of an enhanced activation state, after repeated naïve and cytomix-licensed MSC therapy. These cells consist of monocytes and lymphocytes, but not neutrophils due to exclusion on the flow cytometer, displaying a significant reduction in phagocytic and bacterial killing capabilities after the infection. However, with repeated doses of naïve and cytomix-licensed MSC administrations, it is restored to healthy control levels. Further investigation efforts are needed to elucidate the precise cell populations in which this is occurring.

## 4. Materials and Methods

All preclinical experimentation was approved by the Animal Care in Research Ethics Committee of the National University of Ireland, Galway. Experiments were conducted under an approved licence from the Health Products Regulatory Agency (HPRA), Ireland (AE19125/P067). Specific pathogen-free adult male Sprague-Dawley rats (Envigo, UK) weighing between 350 g and 450 g were used in all experiments.

### 4.1. UC PVC MSCs

Human umbilical cord (hUC) perivascular MSC cell populations were provided by Tissue Regeneration Therapeutics (Toronto, ON, Canada). The UC-hMSCs were thawed and expanded as previously described [43], cultured at 37 °C, 95% humidity, 5% CO_2_ until 70–80% confluent, and then sub-cultured to passage 3–4. At this point, cells were exposed to a cytokine cocktail (cytomix; 50 ng/mL TNF-α, 50 ng/mL IFN-γ, 50 ng/mL IL-1β, all immunotools Ltd.) for 24 h and cryopreserved. Cells were thawed immediately prior to administration, washed in PBS, quantified, and resuspended in 1 mL PBS to yield a dosage of 1 × 10^7^ MSCs/kg. MSC aliquots were excluded from use if the viability was determined to be below 80%.

### 4.2. Preclinical Experimental Series

#### 4.2.1. Pneumonia Induction and Establishment

A live culture of 5 × 10^8^ CFU *K. pneumoniae* was administered in a 300 µL bolus intratracheally (IT) through a 16G intubation tube. Following the procedure, animals were allowed to recover from anaesthesia and rehoused. The animals were monitored using approved distress score sheets at regular intervals, and their status, including behavioural signs and welfare, was recorded for 72 h after pneumonia induction.

#### 4.2.2. Experimental Design

Animals were randomized to receive intravenous (IV) vehicle control or 1 × 10^7^ naïve or cytomix-preactivated UC-MSC/kg 24 h after pneumonia induction while under isoflurane anaesthesia. These animals were further randomised to receive a second 1 × 10^7^ naïve or cytomix-preactivated UC-MSC/kg dose or vehicle control 48 h after pneumonia induction. Animals were rehoused for a further 24 h.

### 4.3. Assessment of Injury and Recovery

#### 4.3.1. Premortem Assessment

At 72 h post pneumonia induction, animals were anaesthetised with subcutaneous ketamine (80 mg/kg, Ketamidor™; Chanelle, Galway, Ireland) and medetomidine (0.5 mg/kg, Medetor™, Chanelle, Galway, Ireland). Surgical tracheostomy was performed to allow mechanical ventilation, and a 22G cannula was inserted in the right carotid artery for blood sampling and monitoring of heart rate and blood pressure throughout the procedure. Anaesthesia was maintained with alfaxalone (2 mg/kg, Alfaxan™; Vetoquinol Ltd., Towcester, UK), and mechanical ventilation was commenced at 7 mL/kg tidal volume. Arterial blood gas analysis was performed (ABL Flex 90, Radiometer, Co. Clare, Ireland), and blood cell counts were obtained using a haemoanalyser (Mythic 18, Orphee, Plan-les-Ouates, Switzerland).

#### 4.3.2. Postmortem Assessment

After exsanguination under anaesthesia, blood and bronchoalveolar lavage (BAL) samples were collected for cytokine profiles and bacterial load measurements. Whole blood was divided and either centrifuged to yield plasma samples or subjected to density gradient separation of peripheral blood mononuclear cells (PBMCs) and granulocytes using histopaque 1077 and 1119, respectively (Both Sigma-Aldrich, Dublin, Ireland).

Bacterial load. BAL fluid and whole blood were plated onto UTI agar plates (Brilliance Clarity, Fannin Ltd., Galway, Ireland) and incubated overnight at 37 °C. The total colony number of each indicative colour was determined.

### 4.4. Inflammatory Cytokine Profile

Cytokine-induced neutrophil chemoattractant (CINC-1), tumour necrosis factor (TNF) α, and interleukin 6 (IL-6) were quantified by ELISA (R&D Systems), and 23 other cytokines and growth factors were measured using a multiplex immunoassay system (Bio-Plex Pro Rat Cytokine, Chemokine and Growth Factor Assay; Bio-Rad Ltd., Watford, UK).

Histological analyses. The left lung lobe was inflated using 4% paraformaldehyde solution and tied off. The lobe was suspended in 4% PFA until use for histological analysis, whereby the lobe was divided into five pieces craniocaudally, processed, and paraffin-embedded. Each lobe piece was sectioned into 7 µm thick cross-sections and stained using haematoxylin and eosin. Stereology was performed on images of each lung section taken at 20× and overlaid with a 10 × 10 grid. The intersections at each 50 um space were scored as tissue, airspace, or non-acinar tissue to determine the percentage airspace in the lung.

### 4.5. Phagocytosis and Superoxide Anion Production

Neutrophil phagocytosis was assessed using pHrodo™ bioparticles opsonised with human serum at 2 mg/mL for 1 h at 37 °C with regular agitation. pHrodo particles were added to the 2.5 × 10^5^ WBCs and BAL cells at 10 particles/cell in a non-tissue culture-treated 96-well plate and incubated for 30 min at 37 °C. Neutrophil superoxide anion production was measured with 80 ng/µL of DHR-123 in complete rat medium and added to 2.5 × 10^5^ WBCs and BAL cells for 30 min at 37 °C in a non-tissue culture-treated 96-well plate. Cells were immediately placed on ice and washed with DPBS containing 10% FBS before being stained with anti-rat RP-1 BB700, CD45 PE, and Live/Dead™ Far Red for 30 min in the dark on ice. Samples were washed and fixed in 4% PFA for 10 min before being analysed on the Accuri C6 flow cytometer. FMOs and appropriate isotype controls were used during the analysis.

Adherent BAL monocyte/macrophages were isolated from BAL and exposed to Zymosan A *S. cerevisiae* BioParticles™ opsonised with human serum 2 mg/mL for 1 h at 37 °C at eight particles per cell along with 0.3 µg/mL DAPI and 0.35 mg/mL NBT solution before being incubated for 30 min. Cells were then fixed with 4% PFA for 10 min before being stored in the dark at 4 °C. Analysis occurred on the Cytation 1 with images taken in brightfield, DAPI, and FITC before being overlayed for counting.

### 4.6. Peripheral Blood and BAL Cell Surface Marker Characterisation

T cell activation was determined by staining 2 × 10^5^ of peripheral and BAL cells using the following anti-rat antibodies: CD3, CD4, CD8, CD25, CD38, and PD-1. PD-1 antibody was only used on BAL samples. Samples were stained in DPBS containing 10% FBS in the dark on ice. Samples were washed before being analysed on the Accuri C6 flow cytometer with FMOs and appropriate isotype controls.

Neutrophil and BAL T cell apoptosis and activation were determined by staining 2 × 10^5^ WBCs and BAL cells with anti-rat RP-1 PE, CD11b/c PE-vio770, and Live/Dead™ Far Red in DPBS containing 10% FBS in the dark on ice. Samples were washed and resuspended in 1× Annexin V binding buffer containing 10% FBS with Annexin V FITC added 10 min before the sample was analysed on the Accuri C6 flow cytometer. FMOs and appropriate isotype controls were used during the analysis.

T cell proliferation assays were performed using 2 × 10^5^ cells which were stained with a 1:500 dilution of CellTrace™ CFSE Cell Proliferation Kit (Invitrogen) in DPBS for 20 min in the dark at room temperature. After the incubation period, the staining was arrested with complete rat T cell medium. Then, 2 × 10^5^ CFSE-labelled cells were added per well of a round-bottom 96-well plate (Sarstedt). To stimulate T cell proliferation, anti-rat CD3 (BD Biosciences) and anti-rat CD28 (BD Biosciences) were added to the wells at 7.5 µg/mL and 2 µg/mL, respectively. Unstained and unstimulated, stained and unstimulated, and stained and stimulated controls were included. Plates were incubated for 4 days at 37 °C, with a medium change occurring after 2 days, after which they were washed and resuspended in DPBS containing 5% FBS. For flow cytometry analysis, the cells were stained with anti-rat CD4 PE-vio770 (Miltenyi), CD8 PE (Miltenyi), and Live/Dead™ before being incubated on ice for 30 min in the dark.

### 4.7. Statistical Analyses

Statistical analysis was performed on scientific data using GraphPad Prism 8.0.1 software. Most results were presented as the mean ± SD. Data were examined for outliers using the ROUT test. Unpaired, two-tailed Student *t*-tests were used to compare relative changes with a significance threshold of *p* < 0.01 to account for multiple comparisons within the sample. One-way ANOVA followed by the Dunnett (to compare all groups to the vehicle control) or Tukey (to compare all groups to each other) method was performed on datasets containing more than two experimental groups, with the significance threshold set at *p* < 0.05.

## 5. Conclusions

In these studies, we demonstrated that a repeated MSC dosage regimen, particularly when preactivated MSCs are administered, enhanced the resolution of established pneumonia and reduced the bacterial load, in a clinically relevant model of established AMR *K. pneumoniae*-induced pneumosepsis. The later stage of infection treated, the efficacy of an enhanced MSC, and the efficacy of a repeated dosing regimen all underline the clinical relevance of the findings, addressing several important hurdles in the translation of MSCs as a therapy for established pneumonia induced by AMR bacteria.

## Figures and Tables

**Figure 1 ijms-24-08055-f001:**
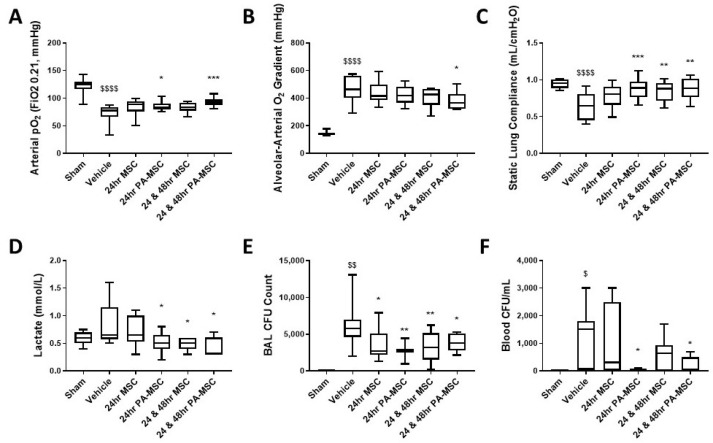
Lung function was significantly improved with the application of MSCs, significantly enhancing arterial pO_2_ with both PA-MSCs (**A**), while only repeated doses significantly improved the alveolar–arterial O_2_ gradient (**B**). Respiratory static compliance was significantly improved with the application of PA-MSCs and repeated doses of MSCs (**C**). The serum lactate was significantly reduced after administration of PA-MSCs and repeated doses of MSCs (**D**). BAL bacterial load was significantly reduced with the application of MSC and PA-MSCs (**E**), while only PA-MSCs significantly reduced blood bacterial load (**F**), with repeated administrations showing no difference. BAL = bronchoalveolar lavage; CFU = colony-forming units; naïve MSC = MSC; cytomix-preactivated MSC = PA-MSC. Box plots and whiskers represent the minimum, first quartile, median, third quartile, and maximum. *, **, *** *p* ≤ 0.05, 0.01, 0.001 versus vehicle control; $, $$, $$$$ = *p* ≤ 0.05, 0.01, 0.0001 versus sham group; *n* = 8–12.

**Figure 2 ijms-24-08055-f002:**
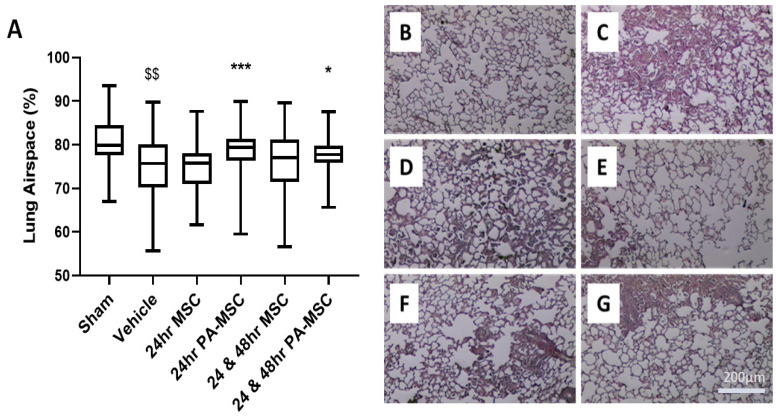
Sections of PFA-fixed, H&E-stained, paraffin-embedded lung tissue were analysed using stereology for the percentage of airspace present in the lungs (**A**–**G**). Animals receiving PA-MSCs showed a significantly higher airspace fraction compared to vehicle control (**A**). Representative images of the corresponding groups are shown at 20× magnification: Sham (**B**), vehicle (**C**), 24 h MSCs (**D**), 24 h PA-MSCs (**E**), 24 and 48 h MSCs (**F**), and 24 and 48h PA-MSCs (**G**). Box plots and whiskers represent the minimum, first quartile, median, third quartile, and maximum. *, *** = *p* ≤ 0.05, 0.001, versus vehicle control; $$, = *p* ≤ 0.01, versus sham group; *n* = 8–12.

**Figure 3 ijms-24-08055-f003:**
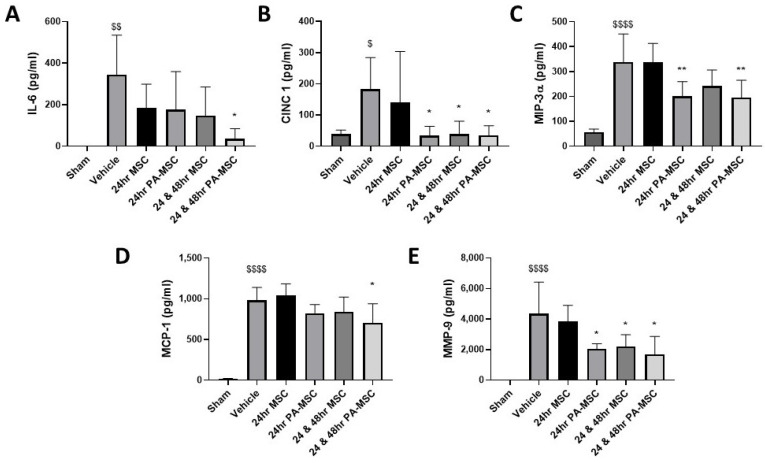
Inflammatory cytokine IL-6 was only reduced in the repeated PA-MSC group (**A**). Chemotaxis agent CINC1 was significantly reduced in the BAL after PA-MSC dose and repeated MSC administration (**B**). MIP-3α was significantly reduced after PA-MSC dose (**C**), while MCP-1 was only significantly reduced after repeated PA-MSC therapy (**D**). MMP-9 was significantly reduced in the BAL after PA-MSC and repeated MSC administration (**E**). IL = interleukin; CINC-1 = cytokine-induced neutrophil chemoattractant 1; MIP = macrophage inflammatory protein; MCP = monocyte chemotactic protein; MMP = matrix metalloprotease. Columns represent the mean; error bars represent the SD. *, **, = *p* ≤ 0.05, 0.01, versus vehicle control; $, $$, $$$$ = *p* ≤ 0.05, 0.01, 0.0001 versus sham group; *n* = 8–12.

**Figure 4 ijms-24-08055-f004:**
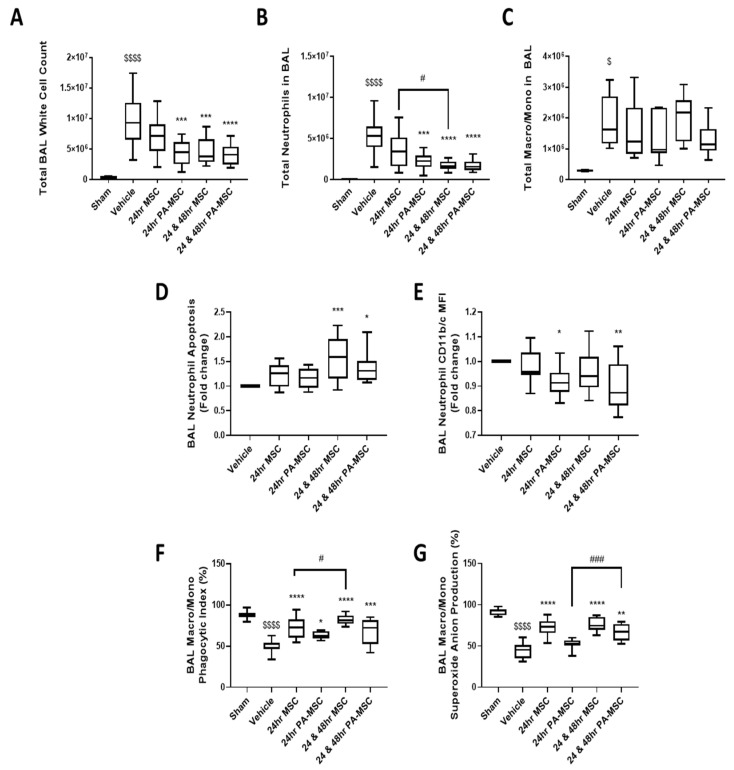
The number of pulmonary white blood cells was significantly elevated in the vehicle control group compared to sham (**A**), whereas PA-MSC- and repeated MSC-treated groups had significantly lower white blood cells in the BAL than vehicle control. This reduction is accounted for by significantly reduced neutrophils for PA-MSC- and repeated MSC-treated groups (**B**). There was no change across all treatment groups for the total number of macrophages/monocytes in the BAL (**C**). Flow cytometry analysis of BAL neutrophils showed that they had a significantly increased rate of apoptosis after repeated MSC and PA-MSC administration (**D**), with the PA-MSC groups significantly reducing their CD11b/c activation state (**E**). Macrophages/monocytes in the BAL had significantly improved phagocytosis across all treatment groups, with repeated doses of MSCs being significantly better than their single-dose counterparts (**F**). Both MSC doses and repeated PA-MSC administration significantly improved macrophage/monocyte superoxide anion production (**G**). BAL = bronchoalveolar lavage. Box plots and whiskers represent the minimum, first quartile, median, third quartile, and maximum. *, **, ***, **** = *p* ≤ 0.05, 0.01, 0.001, 0.0001 versus vehicle control; $, $$$$ = *p* ≤ 0.05, 0.0001 versus sham group; #, ### = *p* ≤ 0.05, 0.001 versus other group; *n* = 8–12.

**Figure 5 ijms-24-08055-f005:**
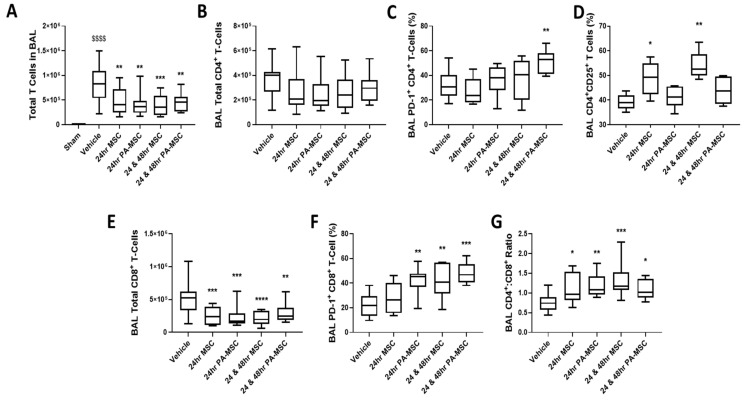
The number of pulmonary T cells was significantly reduced after MSC and PA-MSC treatment (**A**). Flow cytometry analysis of BAL T cells showed no significant difference in the total number of CD4^+^ T helper cells (**B**), but there was a significant increase in the proportion of PD-1^+^ activated T helper cells after repeated PA-MSCs (**C**). MSC administration significantly enhanced the proportion of CD25^+^ T helper cells in the BAL (**D**). There was a significant reduction in the total number of CD8^+^ cytotoxic T cells across all treatment groups (**E**), with PA-MSC dose and repeated MSC treatments significantly increasing the proportion of PD-1^+^ cytotoxic T cells in the BAL (**F**). The CD4/CD8 T cell ratio was significantly increased in the BAL across all treatment groups (**G**). BAL = bronchoalveolar lavage. Box plots and whiskers represent the minimum, first quartile, median, third quartile, and maximum. *, **, ***, **** = *p* ≤ 0.05, 0.01, 0.001, 0.0001 versus vehicle control; $$$$ = *p* ≤ 0.0001 versus sham group; *N* = 8–12.

## Data Availability

Data is available from the authors upon request.

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
