# Peer review of "Multiple Dosing and Preactivation of Mesenchymal Stromal Cells Enhance Efficacy in Established Pneumonia Induced by Antimicrobial-Resistant *Klebsiella pneumoniae* in Rodents"

_ijms, 2023, doi:10.3390/ijms24098055_

Round 1

Reviewer 1 Report

In this study “Multiple dosing and pre-activation of mesenchymal stromal cells enhances efficacy in established pneumonia induced by anti-microbial resistant Klebsiella pneumoniae in rodents” by D. Byrnes et al, the study testing whether repeated MSC and/or preactivation would attenuate Klebsiella pneumonia induced established pneumonia, is based on a previous model in rats published by the same group.

The results presented in here complete their initial results on the role of MSC on such a model of acute inflammation of lungs, but many interpretations of their results must be reviewed, commented, or even removed.

I listed below such concerns.

Major points

1) One of the major points is linked to the previous publication from the same group (Pharmaceuticals 2023, 16,149. https://doi.org/10.3390/ph16020149) comparing the effect of naïve versus cytokine treated MSC with one administration, on the same model of pneumonia induced by anti-microbial resistant Klebsiella pneumoniae in rats.

In the present submitted publication, several effects of one dose treatment (either naïve or PA-MSC) are not similar with those reported within the published publication. The timing of administration, 1h (previous published results) versus 24h  (the present study) might indeed explain such discrepancies. But because the purpose of this submitted study is to compared the multi dosing impact of naïve or PA-MSC, the authors must clearly indicate (more than explained) the different timing between both studies.

I listed below some of the differences between both studies that should be at least commented.

Fig 3A BAL IL-6 > in the contrary to the published study, one administration (24h) with PA-MSC did not reduce the amount of IL6 secreted in the BAL. Explanation for this discrepancy must be provided. How relevant is the result for 24h & 48h PA-MSC treatment?

Similar discrepancies between this study and the published study by the same group are shown for CINC1, MIP3a and MCP1 for the treatment with naïve MSC.

Fig 4G One of the main effects of the 2 doses of PA-MSC treatment (24h/48h) compared to one dose of PA-MSC administration is the restoration of superoxide anion production by the mono/macrophage population in the BAL, one dose being unable to restore this production, while one dose of naïve MSC was.  But in the previous published study, one dose of naïve and one dose of PA-MSC did restore this function. How can be interpreted these 2 different results in the context of multiple dosing? 

Also, in the supplementary figure 1, none of the treatments with one administration (naïve or PA-MSC) did induce production of TNFa, IFNg nor IL1b, while such treatments were shown by the same authors to induce the secretion of such cytokines in the BAL.

2) Analysis of T cells fig 5

In the Fig 5 D, the % of CD4+ CD25+ T cells in the BAL of rats treated with a single or 2 doses of naïve MSC increased significantly. The explanation provided is that the % of Treg cells increased in the BAL of treated animals. CD25 is indeed expressed on CD4 regulatory T cells but is also expressed more generally on all activated CD4 T cells (alpha chain of the IL2 receptor). Because of this, it cannot be considered by itself as a marker for CD4 Treg cells. Expression (or non-expression) of other supplementary markers, such as the absence of CD127 and the expression of FoxP3 should be added to determine the % of CD4 Treg cells. Ideally, a suppressive assay with these CD4 T cells would completely indicate that they are indeed suppressive regulatory T cells.

Therefore, the MSC treatments leading to increased proportions of CD25 CD4 Tcells indicate that either a higher activation of CD4 T cells occurred or that this reflects a higher proportion of CD4 Treg cells, but no data provided can allow to distinguish between both possibilities.

About the CD8 T cells population analyzed in the BAL, the denomination of “memory” CD8 T cells based on the CD25 expression cannot be used. The different populations of CD8 memory T cells (Effector Memory CD8 T cells, Tissue Resident memory CD8 T cells, Central Memory CD8 T cells) are well characterized by several markers other than CD25!

Finally, no information is provided regarding the potential contribution of the administrated human MSC to stimulate the rat T lymphocytes. The increase % of CD25+ CD4+ T cells, PD1+ CD4+ T cells and of PD1+ CD8 T cells observed after MSC administration might be in part due to a xenogenic reaction of rat T cells against human MSC. This should clearly be discussed.

3) In the supplementary section file provided, several supplementary figures and text (from suppl fig 4 to 8) are not explained or presented in the main text.

 All these indications must either be removed or presented and discussed in the main text.

4) In the discussion section (lines 284 to 289), the authors indicate that T cell exhaustion and inactivation (regarding pneumonia infection) can probably be reduced upon multi-doses of MSC administration. This should be taken with precaution. T cell exhaustion is a feature of chronically activated T cells while acute and not chronic infection with Klebsiella pneumoniae are presented in the experiments. Therefore the % of T cells expressing PD1 is more related to an activation state (PD1 is expressed as soon as a T cell is activated).

Again, the denomination of “memory” CD8 T cells based on CD25 expression (CD25+ memory CD8 T cells) is not relevant (see above remarks).

Finally, no demonstration is provided indicating that Treg cell activation is observed  (see comment above)!

Discussion must therefore be changed accordingly, on T cell exhaustion, on “memory” state of T cells and particularly on Treg cells. The sentence “Naïve MSC administration and repeated doses of cytomix licensed MSCs significantly enhance Treg activation state as shown by the increased expression of CD25 on the Tcell surface …” must be removed because it does not reflect what is presented in the study.

Minor points

Line # 143     A short title (“MSC therapy.”), useless, in the middle of the text should be removed.

The legend of suppl fig 2 is untitled Supplementary Figure 3, and this must be corrected.

Author Response

RESPONSE TO REVIEWER 1

Comment 1: One of the major points is linked to the previous publication from the same group (Pharmaceuticals 2023, 16,149. https://doi.org/10.3390/ph16020149) comparing the effect of naïve versus cytokine treated MSC with one administration, on the same model of pneumonia induced by anti-microbial resistant Klebsiella pneumoniae in rats.

In the present submitted publication, several effects of one dose treatment (either naïve or PA-MSC) are not similar with those reported within the published publication. The timing of administration, 1h (previous published results) versus 24h (the present study) might indeed explain such discrepancies. But because the purpose of this submitted study is to compare the multi dosing impact of naïve or PA-MSC, the authors must clearly indicate (more than explained) the different timing between both studies.

I listed below some of the differences between both studies that should be at least commented.

Fig 3A BAL IL-6 > in the contrary to the published study, one administration (24h) with PA-MSC did not reduce the amount of IL6 secreted in the BAL. Explanation for this discrepancy must be provided. How relevant is the result for 24h & 48h PA-MSC treatment? Similar discrepancies between this study and the published study by the same group are shown for CINC1, MIP3a and MCP1 for the treatment with naïve MSC.

Fig 4G One of the main effects of the 2 doses of PA-MSC treatment (24h/48h) compared to one dose of PA-MSC administration is the restoration of superoxide anion production by the mono/macrophage population in the BAL, one dose being unable to restore this production, while one dose of naïve MSC was.  But in the previous published study, one dose of naïve and one dose of PA-MSC did restore this function. How can be interpreted these 2 different results in the context of multiple dosing? 

Also, in the supplementary figure 1, none of the treatments with one administration (naïve or PA-MSC) did induce production of TNFa, IFNg nor IL1b, while such treatments were shown by the same authors to induce the secretion of such cytokines in the BAL.

Response: The reviewer raises an interesting point in regard to differences in findings across 2 of our studies. Two points are important here. Firstly, the data is relatively consistent across the studies, in that MSC therapy lowers the alveolar concentrations of pro-inflammatory cytokines. Second,it is important to understand that aim of the current study is quite different, in that one is testing a more clinically relevant dosing regimen, i.e. administration of first dose MSCs much later in the evolution of the infection, and an examination of the potential to enhance the effect with a repeated dose. In the earlier study, which provided important proof-of-principle for the potential of MSCs in this context, administration very early (i.e. at 1 hour post infection) is likely to abrogate the infection and its immune effects more effectively than later doses. However, this is less clinically relevant, as patients generally present with an established infection rather than earlier in the infection process. Therefore, it is not possible to directly compare the findings across the 2 studies.

We have addressed this important point in the revised discussion. [Lines 284 – 292 and elsewhere in Discussion].

Comment 2: Analysis of T cells fig 5

In the Fig 5 D, the % of CD4+ CD25+ T cells in the BAL of rats treated with a single or 2 doses of naïve MSC increased significantly. The explanation provided is that the % of Treg cells increased in the BAL of treated animals. CD25 is indeed expressed on CD4 regulatory T cells but is also expressed more generally on all activated CD4 T cells (alpha chain of the IL2 receptor). Because of this, it cannot be considered by itself as a marker for CD4 Treg cells. Expression (or non-expression) of other supplementary markers, such as the absence of CD127 and the expression of FoxP3 should be added to determine the % of CD4 Treg cells. Ideally, a suppressive assay with these CD4 T cells would completely indicate that they are indeed suppressive regulatory T cells.

Therefore, the MSC treatments leading to increased proportions of CD25 CD4 T-cells indicate that either a higher activation of CD4 T cells occurred or that this reflects a higher proportion of CD4 Treg cells, but no data provided can allow to distinguish between both possibilities.

About the CD8 T cells population analyzed in the BAL, the denomination of “memory” CD8 T cells based on the CD25 expression cannot be used. The different populations of CD8 memory T cells (Effector Memory CD8 T cells, Tissue Resident memory CD8 T cells, Central Memory CD8 T cells) are well characterized by several markers other than CD25!

Finally, no information is provided regarding the potential contribution of the administrated human MSC to stimulate the rat T lymphocytes. The increase % of CD25+ CD4+ T cells, PD1+ CD4+ T cells and of PD1+ CD8 T cells observed after MSC administration might be in part due to a xenogeneic reaction of rat T cells against human MSC. This should clearly be discussed.

Response: The reviewer raises important points regarding our interpretation of the T cell analyses and we thank you for bringing attention to these issues. The characterization of the CD25+ CD4+ T-cells as T regulatory cells and CD25+ CD8+ T-cells as memory T-cells was based on papers which followed the same nomenclature. However, as you have outlined, it is now an outdated method for identifying these T-cell subpopulations with the recent expansion of rat antibodies being available from reputable sources. This has now been corrected in text to remove any potential confusion regarding their classification.

We have now included and additional panel to Supplementary Figure 7 (E) showing the co-expression of CD25 in CD8+ T-cells to aid in showing an undetectable level of xenogeneic reaction accruing in the animal model as CD25 co-expression on either CD4+ or CD8+ T-cells has been linked to rejection in GvHD. A section in the discussion has been added on the minimal xenogeneic reaction of the human MSCs in rodent models which has been studied extensively in the literature. However, the specific role of human MSCs on rodent PD-1 and CD25 T-cell expression is lacking. Plenty of studies have shown allogenic MSCs suppressing T-cell activation, showing reduced percentage of positive PD-1 and CD25 T-cells and reduced expression of PD-1 on T-cells. As for the specifics of human MSCs on the PD-1 expression on rodent T-cells, we didn’t find any relevant published information in the literature. Since this was something we initially considered in the previously published paper you mentioned, we have included a graph (see attached file) where we did not see any increase in PD-1 positive CD4 and CD8 positive circulating T-cells – while this is 72 hours after the administration you would expect an increased proportion of activated CD4 and CD8 positive T-cells should there have been a xenogeneic reaction against the MSCs. We only did an N of 2, which is why it was not included in the previous publication along with this information not being required by the editors/reviewers, as the percentage of cells was miniscule which showed positivity for the marker.                                                                                                                                                                                   

Comment 3: In the supplementary section file provided, several supplementary figures and text (from suppl fig 4 to 8) are not explained or presented in the main text. All these indications must either be removed or presented and discussed in the main text.

Response: Thank you – we have presented the additional supplemental data description in the main text as we agree with the reviewer that this is important data providing additional insights.                

Comment 4: In the discussion section (lines 284 to 289), the authors indicate that T cell exhaustion and inactivation (regarding pneumonia infection) can probably be reduced upon multi-doses of MSC administration. This should be taken with precaution. T cell exhaustion is a feature of chronically activated T cells while acute and not chronic infection with Klebsiella pneumoniae are presented in the experiments. Therefore the % of T cells expressing PD1 is more related to an activation state (PD1 is expressed as soon as a T cell is activated).

Again, the denomination of “memory” CD8 T cells based on CD25 expression (CD25+ memory CD8 T cells) is not relevant (see above remarks).

Finally, no demonstration is provided indicating that Treg cell activation is observed (see comment above)!

Discussion must therefore be changed accordingly, on T cell exhaustion, on “memory” state of T cells and particularly on Treg cells. The sentence “Naïve MSC administration and repeated doses of cytomix licensed MSCs significantly enhance Treg activation state as shown by the increased expression of CD25 on the T‐cell surface …” must be removed because it does not reflect what is presented in the study.

Response: The reviewer raises additional interesting points regarding our interpretation of the T cell analyses and appreciate you providing your knowledge on the area which can only enhance the work presented here. We have included what we hope is, an appropriate stipulation in regard to T-cell exhaustion in the discussion with a specific mention to chronic infection. We do feel however that the previous point should remain as indices of exhaustion can appear in the early phases of sepsis. Regarding “Naïve MSC administration and repeated doses of cytomix licensed MSCs significantly enhance Treg activation state as shown by the increased expression of CD25 on the T‐cell surface …” this has now been changed to reflect what was actually was presented in the study and thank you for bringing it to our attention.                                                                               

Minor points

Minor Point 1: Line # 143     A short title (“MSC therapy.”), useless, in the middle of the text should be removed.

Response: Thank you – we have corrected this error

Minor Point 1: The legend of suppl fig 2 is untitled Supplementary Figure 3, and this must be corrected.

Response: Thank you for bringing this to our attention – we have corrected this error.

Reviewer 2 Report

Excellent manuscript from my point of view. Repeated dosing and cytomix priming provides increased therapeutic efficacy of MSCs when used directly post cryopreservation thawing in an experimental mouse sepsis model, while a single dose especially of non-cytomix primed MSCs was less effective (often non-significant differences). I would suggest moving the methods section in between introduction and results section for easier accessiblity for the readers.

Author Response

RESPONSE TO REVIEWER 2

Comment 1: Excellent manuscript from my point of view. Repeated dosing and cytomix priming provides increased therapeutic efficacy of MSCs when used directly post cryopreservation thawing in an experimental mouse sepsis model, while a single dose especially of non-cytomix primed MSCs was less effective (often non-significant differences).

Response: Thank you for these positive and constructive comments.         

Comment 2: I would suggest moving the methods section in between introduction and results section for easier accessibility for the readers.

Response: Thank you – we are happy to do this, if the journal agrees, as we are following the prescribed format for IJMS papers.            

Reviewer 3 Report

Dear Authors,

The submitted article entitled " Multiple dosing and preactivation of mesenchymal stromal cells enhances efficacy in established pneumonia induced by anti-microbial resistant Klebsiella pneumoniae in rodents." is a well described study and diserves to be published in the current journal. 

Only minor revisions i have for this work. Below you will find my comments.

1) Please add in figure 2 legend the appropraite scale bar and original magnification of the figures.

2) Please check again the statistics. Epsoecially in those graphs of figures 1-5 where the SD is too high.

Author Response

RESPONSE TO REVIEWER 3

Comment 1: The submitted article entitled " Multiple dosing and preactivation of mesenchymal stromal cells enhances efficacy in established pneumonia induced by anti-microbial resistant Klebsiella pneumoniae in rodents." is a well described study and diserves to be published in the current journal. 

Response: Thank you for these positive and constructive comments.         

Only minor revisions i have for this work. Below you will find my comments.

Comment 2: Please add in figure 2 legend the appropriate scale bar and original magnification of the figures.

Response: Thank you – we have added the scale bar and original magnification as requested

Comment 3: Please check again the statistics. Especially in those graphs of figures 1-5 where the SD is too high.

Response: Thank you – Our box and whisker plots in figures 1,2, 4, 5 depict the minimum and maximum values of that group rather than SD. While some of these datasets have high SD values, we can attribute these to the sample types – e.g. BAL CFU counts generally have a high spread in values which is consistent across series. Our large group numbers enable us to obtain significance despite the variability.

We have rechecked the statistics in the bar graph of figure 3 which has SD values represented by error bars. We concede that the error bars are large in some of the vehicle control groups. This is likely due to the heterogeneity amongst animal models of pneumonia. We did discover however, the statistics for figure 3 panels D and E were incorrect and have been amended accordingly (Panel D, 24&48h PA-MSC = **; Panel E, 24h and 24+48h PA-MSC = **) and we thank the reviewer for enabling this revision.

Round 2

Reviewer 1 Report

My initial reviewing was raising 2 main concerns that the authors adressed in this revised manuscript.

  I would like to thank the authors for the substantial modifications of the interpretations of their results (notably regarding the activation state of T cells after treatment with  MSCs). Their revised interpretation does not remove any interest on the role of these MSC treatments and are more accurate from an immunological point of view.

Regarding the differences between the results reported within this manuscript and within their previous published article, I understand the interst of this study within a context more appropriate for a clinical basis. In my opinion, this study is indeed more relevant, and the results presented here are probably of more interest than that of the first study.